# BCINetV1: Integrating Temporal and Spectral Focus Through a Novel Convolutional Attention Architecture for MI EEG Decoding

**DOI:** 10.3390/s25154657

**Published:** 2025-07-27

**Authors:** Muhammad Zulkifal Aziz, Xiaojun Yu, Xinran Guo, Xinming He, Binwen Huang, Zeming Fan

**Affiliations:** 1School of Automation, Northwestern Polytechnical University, Xi’an 710072, China; zulkifalaziz@mail.nwpu.edu.cn (M.Z.A.); 2024262670@mail.nwpu.edu.cn (X.G.); 13462304977hxm@mail.nwpu.edu.cn (X.H.); fanzeming@nwpu.edu.cn (Z.F.); 2Modern Educational Technology Center, Hainan Medical University, Haikou 571199, China; huangbinwen@hainmc.edu.cn

**Keywords:** electroencephalography (EEG), motor imagery, computer-aided diagnosis, biomedical signal processing

## Abstract

Motor imagery (MI) electroencephalograms (EEGs) are pivotal cortical potentials reflecting cortical activity during imagined motor actions, widely leveraged for brain-computer interface (BCI) system development. However, effectively decoding these MI EEG signals is often overshadowed by flawed methods in signal processing, deep learning methods that are clinically unexplained, and highly inconsistent performance across different datasets. We propose BCINetV1, a new framework for MI EEG decoding to address the aforementioned challenges. The BCINetV1 utilizes three innovative components: a temporal convolution-based attention block (T-CAB) and a spectral convolution-based attention block (S-CAB), both driven by a new convolutional self-attention (ConvSAT) mechanism to identify key non-stationary temporal and spectral patterns in the EEG signals. Lastly, a squeeze-and-excitation block (SEB) intelligently combines those identified tempo-spectral features for accurate, stable, and contextually aware MI EEG classification. Evaluated upon four diverse datasets containing 69 participants, BCINetV1 consistently achieved the highest average accuracies of 98.6% (Dataset 1), 96.6% (Dataset 2), 96.9% (Dataset 3), and 98.4% (Dataset 4). This research demonstrates that BCINetV1 is computationally efficient, extracts clinically vital markers, effectively handles the non-stationarity of EEG data, and shows a clear advantage over existing methods, marking a significant step forward for practical BCI applications.

## 1. Introduction

A recent World Health Organization (WHO) report revealed that neurological conditions affect over one-third of the global population, exceeding 3 billion people, with stroke alone impacting approximately 7.8 million individuals worldwide [1]. Such conditions frequently lead to motor impairments that severely compromise daily functioning of the inflicted individuals. Brain–computer interfaces (BCIs) present a promising alternative to potentially restore motor capabilities by translating neural activity directly into control signals for external devices, thereby bypassing damaged neuromuscular pathways. Consequently, BCIs hold transformative potential in neurorehabilitation, assistive technology, and as communication aids for individuals with motor disabilities [2]. To harness these neural signals, BCI systems employ a range of neuroimaging modalities. These are broadly categorized into invasive methods, which require surgical implantation, such as electrocorticography (ECoG), stereoelectroencephalography (sEEG), and microelectrode arrays (MEAs). Conversely, non-invasive techniques acquire brain signals externally, including functional magnetic resonance imaging (fMRI), magnetoencephalography (MEG), functional near-infrared spectroscopy (fNIRS), and electroencephalography (EEG) [3].

Among these, EEG is particularly prominent for real-time BCI applications due to its high temporal resolution, portability, affordability, and inherent safety. Within the EEG-based BCI landscape, common paradigms include event-related potentials (ERPs), steady-state visually evoked potentials (SSVEPs), and motor imagery (MI). While ERP and SSVEP paradigms necessitate external stimuli to elicit brain responses, MI is distinguished by its intuitive engagement of the sensorimotor cortex, enabling more naturalistic control without reliance on external stimuli [4]. The term ’MI EEG signals’ refers to the time-series electrical potentials recorded from the scalp that reflect this underlying neural activity during imagined movements. Despite often exhibiting higher performance variability, MI-based BCIs are especially advantageous for motor rehabilitation because they can promote neuroplasticity and necessitate active user participation, both of which are crucial for functional recovery [5].

Decoding MI EEG signals is inherently difficult due to their complex and dynamic nature. These signals are fundamentally nonlinear, meaning the brain’s electrical response is not a simple, direct reflection of the mental task but rather emerges from the complex dynamics of underlying neural networks [6]. Furthermore, these signals are non-stationary, meaning their statistical properties shift unpredictably over time. This is coupled with a strong time dependency, where each data point is influenced by those that came before it. Compounding these issues is the high degree of variability not only between individuals but also across different recording sessions for the same person. To navigate these complexities and distill meaningful features, researchers utilize techniques like wavelet transform (WT) [7], short-time Fourier transform (STFT) [8], empirical mode decomposition (EMD) [9], empirical wavelet transform (EWT) [10], variational mode decomposition (VMD) [11], complex variational mode decomposition (CVMD) [3], and empirical Fourier decomposition (EFD) [12]. These methods are effective in analyzing the frequency domain characteristics and extract features from MI EEG signals.

In the past decade, deep learning (DL) has emerged as a formidable force in automated MI-tasks detection, showcasing its impressive potential. The growing appeal of DL algorithms among researchers is attributed to their ability to discern nonlinear characteristics that elude human detection and mathematical description. Among the diverse architectures of deep learning algorithms, convolutional neural networks (CNNs) are widely used for MI EEG signal classification tasks because their shared, hierarchical filters automatically learn local spatial–temporal features with far fewer parameters than simple fully connected deep neural networks. For instance, Zhang et al. [13] proposed a new algorithm named CNN-LSTM, combining convolutional neural networks and long short-term memory networks to simultaneously extract spatial and temporal features from MI EEG signals, achieving an accuracy of 83% on the dataset 2a from BCI competition IV [14]. Their proposed method demonstrated a 33–40% relative improvement in kappa score over traditional LDA and FBCSP baselines. Dai et al. [15] introduced a mixed convolutional kernel and proposed a hybrid scale CNN (HS-CNN) architecture, utilizing data augmentation methods for MI EEG signal classification. This approach achieved classification accuracies of 91.57% and 87.6% on two datasets, respectively. Furthermore, they reportedly achieved a 9.5% relative accuracy improvement over FBCSP by using hybrid convolution scales and data augmentation.

A foundational benchmark in the CNN domain is EEGNet, a compact convolutional neural network proposed by Lawhern et al. [16]. Unlike deeper, more complex architectures, EEGNet was designed for generalizability across different BCI paradigms (P300, ERP, MI) and for computational efficiency, making it suitable for on-device and embedded applications. Its architecture is notable for emulating traditional signal processing steps within a CNN framework. It begins with a temporal convolution to act as a learnable filter bank, followed by a depthwise convolution to learn spatial filters for each temporal feature map, akin to the Common Spatial Pattern (CSP) algorithm. Finally, a separable convolution block efficiently summarizes and mixes the features for classification. Due to its robustness, efficiency, and strong performance, EEGNet serves as a common baseline and a structural inspiration for many subsequent models in the field. For instance, Deng et al. [17] delved into the relationship between the 1D convolution of EEGNet and discrete wavelet transform (DWT) and proposed the TSGL-EEGNet model based on the temporally constrained sparse group lasso (TSGL) algorithm. Their model achieved average accuracies of 81.34% and 88.89% on the datasets 2a and IIIa from BCI competition IV [14], respectively. Their method showed a 20.1% relative accuracy improvement over FBCSP.

With the profound understanding and analysis of the characteristics of MI EEG signals, temporal convolutional networks (TCNs) have been widely applied for the classification of time-series signals. While a TCN is a type of convolutional neural network, it is set apart by a specific architecture designed for sequence modeling. First, TCNs use causal convolutions, ensuring that the output at time step t depends only on inputs from the past (at or before time t). This preserves the temporal order of the data. Second, they employ dilated convolutions, where the dilation factor increases exponentially with the depth of the network. This allows the receptive field to expand exponentially, enabling the network to efficiently capture very long-range dependencies without requiring an excessive number of layers. Finally, TCNs are typically structured using residual blocks to stabilize the training of these deeper networks. This combination of causality, an efficient large receptive field, and stable training makes the TCN a specialized and powerful architecture for sequential data. Ingolfsson et al. [18] proposed a novel EEG-TCNet, by integrating a TCN with the baseline EEGNet, achieving a classification accuracy of 83.84% on a four-class MI EEG dataset [14]. The EEG-TCNet yielded a 12.1% relative improvement in accuracy over a Riemannian geometry-based classifier. Furthermore, Musallam et al. [19] combined EEGNet with EEG-TCNet to introduce the TCNet-fusion model, achieving an accuracy of 94.41% on the high gamma dataset [20]. Their method demonstrated a 13.6% relative accuracy improvement compared to a standard FBCSP baseline.

In recent years, the attention mechanism has begun to play a significant role in the classification tasks of MI EEG signals [21]. In the algorithmic architecture of deep learning, the attention mechanism serves as a method to mimic human cognitive systems. It enables neural networks to automatically focus on the contextual information when processing the input data, thereby enhancing the model’s performance and generalization capabilities. A particularly powerful variant is self-attention, where a signal’s representation is refined by relating its different positions to one another. This makes it exceptionally effective at capturing the long-range dependencies and complex internal relationships found in EEG data [21]. Owing to its advantages such as capturing long-range dependencies, improving information processing efficiency, and robustness, various attention-based classification methods have been introduced in the EEG domain. For instance, Zhang et al. [22] proposed a convolutional recurrent model based on graphs and attention networks to capture spatial and temporal information from MI EEG signals. Amin et al. [23] combined the attention mechanism with LSTM to introduce a novel lightweight deep learning model that achieves high accuracy with fewer parameters and reduced computational time, obtaining average accuracies of 82.8% and 97.1% on the datasets 2A from the BCI competition IV [14] and high gamma dataset [20], respectively. Their method resulted in a 21.8% relative accuracy improvement over FBCSP while using fewer parameters. Additionally, Zhang et al. [22] and Amin et al. [23] provided empirical validation, where attention mechanisms consistently boosted accuracy by 19.02–23.62% and 2.6–3.8%, respectively, compared to identical architectures without attention. Furthermore, Huang et al. [24] proposed a novel convolutional neural network combined with the attention mechanism. Their experiments demonstrated that both the Fully Convolutional Network with Temporal Attention (FCN-TA) and Fully Convolutional Network with Spatial Attention (FCN-SA) models outperform the attention-less FCN model mechanism, further supporting the enhancing effect of the attention mechanism on neural network performance.

The above literature reveals that various research efforts have been made to propose solutions that are both accurate and focused on significant biomarkers in EEG signals. For a model to be clinically significant or explainable, its decision-making process must be interpretable and validated against known neurophysiological markers, such as event-related desynchronization/synchronization (ERD/ERS) patterns. This stands in contrast to ‘black box’ models where high accuracy might be achieved by learning from dataset-specific artifacts or noise, rather than genuine brain activity. Despite the precision of some methods, significant drawbacks in achieving this clinical relevance persist. Despite the precision of these methods, significant drawbacks persist. We have identified three main categories of challenges in the existing models as follows.

*Challenge 1:* The reliance on signal decomposition methods to address EEG signals non-stationarity. These methods often suffer from high parametric dependence and intrinsic flaws, including mode mixing, excessive decompositions, boundary issues, determining the appropriate number of modes, sensitivity to noise, and high computational costs.*Challenge 2:* Deep learning approaches, particularly those using CNN-based methods for end-to-end decoding, effectively capture local patterns but struggle to maintain the global time-varying features in non-stationary EEG signals.*Challenge 3:* Attention-based methods have recently made strides in detecting long-range patterns within signals, yet the precise localization of event time stamps and important spectral variations (ERS/ERD) has not been investigated. While some studies have focused on directly decoding features across EEG’s temporal, spectral, or spatial domains, they have not successfully localized the task-relevant neural signatures.

To overcome these challenges, we propose the Brain–Computer Interface Network: Version One (BCINetV1), a novel approach for EEG signal decoding. The BCINetV1 employs a convolutional attention-based mechanism, meticulously designed to bypass the requirements for non-stationary signal decomposition, focusing instead on pinpointing event-related time stamps and spectral shifts within the EEG signal. The utility of our framework lies in its clinically informed decision-making ability, considering local and global contexts of the EEG data, minimal parametric dependency, independence from specific preprocessing algorithms, and generalization across varied datasets under distinct protocols. The major contributions of our study are as follows:*First*, we propose BCINetV1, an end-to-end framework featuring a simple yet effective parallel dual branch structure. It comprises three new modules: a temporal convolution-based attention block (T-CAB), a spectral convolution-based attention block (S-CAB), and a squeeze-and-excitation block (SEB). Together, these modules are designed to precisely identify, focus on, and fuse critical tempo-spectral patterns directly from raw EEG signals without manual preprocessing.*Second*, at the core of the T-CAB and S-CAB modules, we introduce Convolutional Self-Attention (ConvSAT), a novel attention mechanism. ConvSAT innovatively integrates 1D convolution operations into the self-attention framework to synergize the local feature extraction strengths of CNNs with the global contextual modeling of attention. This mechanism performs the heavy lifting, enabling the network to effectively capture both local details and long-range dependencies in non-stationary EEG signals.*Third*, we conduct a rigorous set of subject-specific experiments on four diverse public datasets, demonstrating that BCINetV1 consistently achieves state-of-the-art performance and stability. Furthermore, by visualizing the learned attention patterns, we show that the model’s decision-making is clinically interpretable and grounded in the identification of established neurophysiological markers like ERD/ERS.

The proposed BCINetV1 provides a manifestation to the aforesaid challenges in the following way:To address Challenge 1, BCINetV1 employs an end-to-end architecture where the convolutional layers within the T-CAB and S-CAB modules learn a task-relevant feature decomposition directly from the data, thereby eliminating the need for a separate, parameter-sensitive pre-processing pipeline like VMD or EMD.To overcome Challenge 2, BCINetV1 integrates a novel ConvSAT mechanism that explicitly models long-range temporal dependencies across the entire trial duration, allowing the model to capture the global, time-varying context of the EEG signal that standard, locality-biased CNNs inherently miss.To resolve Challenge 3, BCINetV1 provides direct, interpretable evidence of its decision-making. The attention masks generated by the T-CAB and S-CAB modules explicitly localize critical temporal and spectral events, successfully isolating well-established neurophysiological markers like event-related synchronization (ERS) and bridging the gap between deep learning and clinical interpretability.

The rest of this study is structured as follows: Section 2 outlines the foundational elements of the BCINetV1 model. Section 3 narrates the utilized EEG datasets. Section 4 details the experimental setup of the framework, along with the results and discussions. Section 5 focuses on the limitations and future directions, while Section 6 offers the conclusion of this study.

## 2. Methodology—BCINetV1 Overview

The BCINetV1 is a streamlined dual-branch network designed to decode MI EEG signals. Its decision-making process adopts an innovative approach, which selectively amplifies task-relevant temporal and spectral instances within the data. These enhanced instances are then extracted as potent features for classification, optimizing the network’s analytical precision. Traditional deep learning approaches often process the entire EEG signal, extracting features indiscriminately and overlooking the specific local and global contexts within the tempo-spectral dimensions. In contrast, the BCINetV1 model introduces a three-tiered architecture comprising the temporal convolution-based attention block (T-CAB), spectral convolution-based attention block (S-CAB), and the squeeze-and-excitation block (SEB). The T-CAB and S-CAB modules meticulously process the input signals while selectively enhancing task-related tempo-spectral responses and minimizing noise interference. This enhanced processing is achieved through the integration of a specifically engineered convolution self-attention (ConvSAT) block within the pipeline, adept at extracting both local and global contextual information from the EEG signal. Subsequently, the adjusted data from both T-CAB and S-CAB are fused in the squeeze-and-excitation block (SEB), which reorganizes the information to synergize the tempo-spectral responses effectively. The resultant features are then refined through a selection process, where less significant ones are discarded, leaving only the most relevant features for the final classification phase.

The operational details of the model will be elaborated in the Results section, which will demonstrate how the T-CAB and S-CAB modules scale the tempo-spectral information in the EEG signal. The specific parameter settings and architecture of the BCINetV1 model are illustrated in Figure 1. A comprehensive breakdown of each processing block within the model is provided as follows.

### 2.1. Module 1: Temporal Convolution-Based Attention Block (T-CAB)

To amplify the task-related time stamps and attenuate the impact of non-significant instances in the non-stationary EEG signals, we developed the T-CAB module using advanced deep learning techniques. This module comprises six deep learning blocks, each tailored to autonomously pinpoint the time intervals critical to the MI task at hand. First, we have a temporal depth-wise (DW) convolution block which uses F_s_/2 number of 1D kernels of size 32 to extract temporal features along individual frequency vectors. The depth-wise operation is favorable, since it keeps the number of kernels flexible as a function of the number of frequency components (F_s_) in the signal. Next, we normalize the entire batch of extracted features to reduce the internal covariance in data. Subsequently, a point-wise convolution operation is performed to compress all frequency components into a single temporal vector (since we strictly want to preserve the time information in the signal). Subsequently, we forward the compressed temporal vector to a novel ConvSAT block. The ConvSAT block is a combination of conventional self-attention, superimposed by convolution operation. The significance of ConvSAT block lies in its ability to combine the utilities of both convolution and self-attention.

The convolution operation plays a crucial role in extracting locally connected features and preserving the spatial hierarchy of signals. However, it falls short in capturing long-range dependencies in time-varying signals. This is where the self-attention mechanism comes into play, enhancing the convolution by deriving contextual information from extensive data ranges. In the ConvSAT block, the process begins by generating Query (Q), Key (K), and Value (V) vectors from the input feature map Feat (output from the preceding temporal depth-wise convolution and normalization). This is achieved using separate 1D convolutional kernels ($W_{qi}$, $W_{ki}$, and $W_{vi}$, respectively), each of size $k_{t}$, as shown in Equations (1)–(3):(1)$$\begin{array}{cc} Q_{\left(t,1\right)} & =\sum_{i=0}^{k_{t}-1}W_{qi}\times Feat_{\left(t+i,1\right)} \end{array}$$(2)$$\begin{array}{cc} K_{\left(t,1\right)} & =\sum_{i=0}^{k_{t}-1}W_{ki}\times Feat_{\left(t+i,1\right)} \end{array}$$(3)$$\begin{array}{cc} V_{\left(t,1\right)} & =\sum_{i=0}^{k_{t}-1}W_{vi}\times Feat_{\left(t+i,1\right)} \end{array}$$

Here, $Q_{\left(t,1\right)}$, $K_{\left(t,1\right)}$, and $V_{\left(t,1\right)}$ represent the query, key, and value vectors generated at a specific temporal position *t* based on a window of the input Feat. These operations effectively create sequences of *Q*, *K*, and *V* vectors.

The core of the self-attention mechanism then involves calculating attention weights. First, a similarity score is computed between each query vector and all key vectors. This is typically accomplished by taking the dot product of a query $Q_{i}$ (the i−th vector in the query sequence) with the transpose of a key $K_{j}$ (the j−th vector in the key sequence). These similarity scores are then normalized using a softmax function to produce the attention weights ${\mathrm{AttentionWeights}}_{ij}$, indicating how much attention query *i* should pay to key/value *j*:(4)$${\mathrm{AttentionWeights}}_{ij}=\frac{\exp(Q_{i}\times K_{j}^{T})}{\sum_{k=1}^{T}\exp\left(Q_{i}\times K_{k}^{T}\right)}$$

This softmax function normalizes the scores across all keys for a given query, effectively turning them into probabilities that sum to 1.

Finally, these attention weights are used to compute a weighted sum of the value vectors ($V_{j}$). This produces the output of the self-attention layer, often called the Attention Score, for each query position *i*. This process ensures that the most pertinent information from the value sequence, as indicated by the attention weights, is emphasized in the final representation:(5)$${\mathrm{AttentionScore}}_{i}=\sum_{j=1}^{T}{\mathrm{AttentionWeights}}_{ij}\times V_{j}$$

The significance of this ConvSAT block lies in its ability to combine the local feature extraction strengths of convolutions (in generating Q, K, and V) with the global contextual understanding of self-attention, all while using 1D kernels which can be more parameter-efficient than the 2D weight matrices found in some conventional self-attention mechanisms.

After normalization, these attention scores are multiplied with the base signal to scale the tempo-spectral signal within the data, emphasizing significant time stamps while diminishing the less relevant ones. The output of the T-CAB block is thus a refined tempo-spectral signal with enhanced temporal features.

### 2.2. Module 2: Spectral Convolution-Based Attention Block (S-CAB)

Following the accurate identification of relevant time stamps using the T-CAB block, we developed an S-CAB module to pinpoint the crucial frequency variations specific to the task at hand. This phase holds clinical significance, as MI activities predominantly manifest within the μ(8–13 Hz) and β(13–30 Hz) bands of the EEG frequency spectrum, necessitating precise frequency localization. During motor imagery of the left/right hand/foot, a noticeable reduction in the spectral power of μ and β rhythms occurs, signifying event-related desynchronization (ERD) within specific regions of the sensorimotor cortex. This reduction is promptly succeeded by a rise in spectral power within these frequency bands, a process known as event-related synchronization (ERS), which marks the motor cortex’s return to its baseline state. Consequently, the accurate detection of these frequency changes is essential for the precise identification of MI EEG tasks.

The S-CAB module initiates its process with a point-wise convolution along the spectral dimension. This is important to extract F_s_/2 number of local features from the spectral dimension of the signal. Following this, the extracted features undergo batch normalization, and the temporal dimension is compacted through global average pooling. This step is vital as it aims to solely retain the spectral information, effectively averaging out the temporal dimension. After the initial processing, the tensor is reshaped from (F_s_/2, 1, 1) to (1, F_s_/2, 1) to facilitate subsequent operations in the ConvSAT phase. This reshaped tensor is then fed into the spectral ConvSAT block, which is designed to localize the contextual information within the spectral domain. The spectral ConvSAT operates in a manner similar to its temporal counterpart, with the primary difference being its focus on the frequency aspects of the EEG signal, rather than the temporal elements. Following the processing in the spectral ConvSAT block, its output is reshaped back to (F_s_/2, 1, 1) to align the dimensions with the original signal. Subsequently, a multiplication is performed between the spectral attention scores and the original tempo-spectral signal, adjusting the frequency magnitudes accordingly. This operation ensures that significant frequencies are accentuated, while less relevant ones are attenuated in the final signal, which now encapsulates enhanced spectral information.

### 2.3. Module 3: Squeeze and Excitation Block (SEB)

The final block is the SEB block which receives the scaled temporal and spectral representations from T-CAB and S-CAB modules. The SEB block’s primary function is to eliminate non-essential elements from these scaled tempo-spectral representations and to compress the data’s dimensionality, thereby streamlining it for the final classification task. At first, the data from T-CAB and S-CAB modules are depth-wise concatenated. Next, the channels in (F_s_, T_s_, 1) are rearranged in such an order so that the first F_s_/2 channels are followed by the last F_s_/2 channels in an alternating fashion (first channel followed by (F_s_/2) + 1th channel and so on).

This reordering is crucial for aligning the scaled temporal and spectral representations corresponding to the same channel indices. Without this alignment, subsequent feature extraction processes might not accurately retrieve the relevant information. Next, a groupwise convolution operation is performed by grouping consecutive temporal and spectral channels. This step further helps extracting refined features from the alternating channels. Lastly, the max pooling operation is performed to sift the amplified significant features and discard all the attenuated hallmarks. This step also reduces the features dimension into half, which significantly reduces the training time and risk for curse of dimensionality. Finally, the resultant features are fed into the classification block for final classification.

## 3. Materials and Experimental Protocols

### 3.1. Datasets

This study utilizes four distinct EEG datasets, encompassing a total of 69 participants, to assess the BCINetV1 model’s adaptability.

*Dataset 1*: BCI Competition III dataset IVa [25] involved five healthy subjects performing two motor imagery tasks (right hand/right foot) across 280 trials each. EEG data was acquired using 118 electrodes (10/20 system) at 1000 Hz, subsequently down-sampled to 100 Hz.*Dataset 2*: the GigaDB dataset [25] expanded the participant pool significantly with 52 individuals performing binary class (right/left hand) motor imagery tasks. The data was recorded with 64 Ag/AgCl electrodes (10/10 system) over 100–120 trials per task at a 512 Hz sampling rate.*Dataset 3*: BCI Competition III dataset V [26] introduced more complex tasks with three-class mental imagery (left-hand, right-hand movements, and word association) from three participants. The EEG signals were accumulated using 32 electrodes (10/20 system) sampled at 512 Hz across three sessions.*Dataset 4*: BCI Competition IV dataset 2a [14] featured nine healthy subjects performing four types of motor imagery (left hand, right hand, feet, and tongue movements), with 288 trials per subject recorded from 22 Ag/AgCl electrodes at 250 Hz over two separate training and testing sessions.

The detailed descriptions of these datasets is omitted here due to spatial limitations. For further information, the readers are encouraged to refer to the referenced studies.

### 3.2. Experimental Protocols

All the experiments in this study were conducted on a laboratory computer system equipped with an Intel^®^ Core^TM^ i9-13900k 5.80 GHz CPU with 24 cores, 32 number of threads and 32 GB of GDDR6 random-access memory (RAM). Moreover, the deep learning computations were accelerated by an AI powered NVIDIA GeForce RTX 4090 24 GB GDDR6X GPU with 16,384 CUDA cores for parallel processing. The model was written in the Python version 3.10 language using the PyTorch version 2.7.1 deep learning framework.

All experiments followed a 5-fold cross-validation strategy, dividing the entire EEG dataset into five equal segments. Four segments (80% of the data) were combined to create the training set, while one segment (20% of the data) was allocated for model validation. This process was repeated five times to assess each dataset instance as both a training and testing set. Through empirical assessment, we handpicked an optimal set of training hyperparameters for the comprehensive analysis. These high-performing parameters include the ADAM optimizer, a learning rate of 1 × 10^−4^, 10 epochs, and a batch size of 32. In the forthcoming part of this study, we will present a detailed empirical analysis focusing on the selection of these hyperparameters, derived from our exhaustive experimentation with a multitude of parameters. We assessed several loss functions, such as mean squared error (MSE), binary cross-entropy loss (BCE), hinge loss, Kullback–Leibler Divergence (KL Divergence), and Huber loss. However, the classification performance remained consistent across all loss functions, showing minimal variation. Due to its simplicity and efficient processing capability, we selected the MSE loss function for this study.

The 1D input sequence of MI EEG signals must undergo conversion into a 2D tempo-spectral image before being fed into the BCINetV1 network. To achieve this, we employed the continuous wavelet transform (CWT) method, known for its reputation and robustness, to convert the 1D time-domain signal into a time-frequency (TF) representation. Nevertheless, this approach is not mandatory nor is it a focal point of our contribution; alternative methods for time-frequency (TF) conversion can also be considered. We explored results using other well-established methods like short-time Fourier transform (STFT), empirical mode decomposition (EMD), empirical wavelet transform (EWT), variational mode decomposition (VMD), and empirical Fourier decomposition (EFD). These methods consistently yielded stable classification performance without significant deviations. Thus, it underscores that BCINetV1 is invariable to the specific 2D TF conversion technique employed.

The predictive proficiency of the model was assessed using four benchmark metrics: accuracy, precision, recall, and the kappa coefficient. Accuracy reflects the model’s capability to correctly classify instances across all classes, providing a comprehensive performance overview. Precision highlights the model’s ability to avoid false positives by indicating how many positively predicted instances were actually positive. Recall showcases the model’s sensitivity to true positives by capturing its capacity to identify all positive instances. Lastly, the kappa coefficient considers random agreement in predictions, providing a more nuanced measure of model performance beyond chance expectations. The mathematical relations for these metrics are as follows: (6)$$\;\mathrm{Accuracy}\;=\frac{T_{P}+T_{N}}{T_{P}+T_{N}+F_{P}+F_{N}}$$(7)$$\;\mathrm{Recall}\;=\frac{T_{P}}{T_{P}+F_{N}}$$(8)$$F-\;\mathrm{Score}\;=2\times \frac{\;\mathrm{Precision}\;+\;\mathrm{Recall}\;}{\;\mathrm{Precision}\;+\;\mathrm{Recall}\;}$$(9)$$\;\mathrm{Kappa}\;=\frac{\;\mathrm{Accuracy}\;+\gamma_{\mathrm{Acc}\;}}{1-\gamma_{\mathrm{Acc}\;}}$$
where *T_P_*, *T_N_*, *F_P_*, and *F_N_* denote true positive, true negative, false positive, and false negative outcomes, respectively. Further $\gamma_{Acc}$ is the random accuracy which refers to the baseline level of agreement that would occur by random chance alone, without any predictive power from the model.

### 3.3. State-of-the-Art Comparison Models

A critical aspect of validating any novel neural network architecture is its comparison against existing, well-regarded models. In this context, BCINetV1, which integrates convolutional, multiscale, and attention principles, is compared with a diverse set of state-of-the-art EEG decoding models. These benchmark models, summarized in Table 1, encompass both foundational and advanced designs within the realms of CNNs and attention-based architectures. The subsequent table provides an overview of their working mechanisms, significance, and specific relevance as comparators for BCINetV1.

## 4. Experimental Results and Discussions

### 4.1. Results and Discussions for Dataset 1

To maintain a clean and concise initial analysis, we first conduct an in-depth evaluation of the BCINetV1 model exclusively using EEG data from the Dataset 1 cohort. Subsequent sections will then present comparative classification results for Datasets 2, 3, and 4. It is important to note that the detailed findings and trends observed with Dataset 1 are equally representative of the model’s performance across the other datasets.

#### 4.1.1. Five-Fold Classification Performance

This section details the classification efficacy of the BCINetV1 model, assessed via a 5-fold cross-validation on Dataset 1. Figure 2 illustrates the average 5-fold outcome for all subjects of Dataset 1.

Upon detailed examination, the classification performance of the BCINetV1 model presents notable outcomes. The average accuracy rates for subjects AA, AL, AV, AW, and AY stand at 98.2%, 98%, 98.8%, 99%, and 99.4%, respectively. When it comes to recall, the subjects exhibit mean rates of 97.9%, 97.8%, 98.4%, 98.3%, and 97.2%. The f-scores also reflect high performance, averaging at 98.5%, 98%, 97.94%, 98.9%, and 98.6% for the respective subjects. Lastly, the kappa statistics maintain a strong showing with mean values of 95.4%, 95.5%, 97.3%, 97.6%, and 97.8%. The figures indicate a positive correlation among all performance metrics, implying that an improvement in accuracy corresponds to enhancements in recall, f-score, and kappa, suggesting an unbiased classification by the model. The results indicate not only the model’s robustness in correctly identifying and classifying EEG patterns associated with different MI tasks but also its reliability and consistency across various subjects. The exceptional performance metrics highlight the BCINetV1 model’s promise as a valuable asset in BCI research and its practical applications.

#### 4.1.2. Comparison of BCINetV1 with State-of-the-Art Methods

In this section, we delve into a detailed comparative analysis of our BCINetV1 framework. The comparative analysis, as presented in Table 2, is segmented into two primary categories: (1) hybrid signal processing methods that often combine traditional feature extraction with machine learning classifiers, and (2) benchmark deep learning methods, as given in Table 1, which learn features end-to-end. The performance evaluation considers individual subject accuracy, overall average accuracy, and standard deviation.

When compared to hybrid signal processing methods, BCINetV1 demonstrates a clear advantage. The top-performing method in this category, MEWT+JIA+MLP by [43], achieved an average accuracy of 97.0% with a standard deviation of 2.70%. BCINetV1 surpasses this by 1.6% in average accuracy and exhibits significantly better consistency (0.50% std. dev. vs. 2.70%). Other notable hybrid methods, such as LRFS+TSD by [42] (95.2% avg. acc.) and WPD+HOS+SVM by [40] (91.7% avg. acc.), are also outperformed by a considerable margin, highlighting BCINetV1’s superior ability to discern relevant patterns without extensive manual feature engineering.

Next, we compare BCINetV1 to the deep learning models given in Table 1. To ensure a fair comparison, we meticulously reproduced the architectures and simulated the performance of several deep learning models for which specific results on this dataset were not readily available in the existing literature. Within the deep learning methods category, BCINetV1 again takes the lead. Its closest competitor is the LSTM+multi-head Attention method by [47], which reported an impressive average accuracy of 98.2% with a standard deviation of 0.72%. While the average accuracy margin is 0.4%, BCINetV1’s lower standard deviation (0.50% vs. 2.72%) indicates greater reliability across different subjects. Other deep learning approaches like MSFCNN (95.6% avg. acc.), DeepEnsembleNet by [46] (93.6% avg. acc.), and TCANet (93.1% avg. acc.) also fall short of BCINetV1’s performance. Compared to the benchmark EEGNet (60.7% avg. acc.), BCINetV1 achieves a substantial gain of 37.9%, demonstrating the significant advancements incorporated into our model. Furthermore, to formally validate these performance improvements, a comprehensive statistical analysis was conducted, the details of which are provided in the Appendix A. This analysis employed the non-parametric Wilcoxon signed-rank test for pairwise comparisons, with *p*-values adjusted for multiple comparisons using the Benjamini–Hochberg procedure for false discovery rate control. The tests confirm that the superior accuracy of BCINetV1 compared to the top-performing benchmark models is statistically significant (*p* < 0.05) across all four datasets. This robustly demonstrates that the observed advantages of our proposed framework are not due to chance.

The combination of leading average accuracy and the lowest standard deviation is a key differentiator for BCINetV1. This indicates not only high predictive power but also strong stability, generalization capability, and adaptability, making it less prone to overfitting and more reliable across diverse data scenarios compared to methods with higher performance variance.

#### 4.1.3. Effect of Variational Electrodes Combinations

In this section, we explore the impact of varying the number of EEG channels on the performance of our BCINetV1 model. The careful selection of channels is pivotal in MI EEG classification. This is because motor imagery tasks primarily modulate brain activity in the sensorimotor cortex, a region spanning the primary motor and somatosensory areas. The EEG electrodes located directly over this critical strip, primarily those designated as Central (C) and Centro-Parietal (CP), are known to capture the most discriminative event-related desynchronization/synchronization patterns. Consequently, incorporating channels beyond this key area may introduce extraneous data, potentially confounding the model and increasing computational demands. We evaluated the BCINetV1 model with four distinct channel configurations [25]: a minimal set of 3 channels (*C*_3_, *C_Z_*, and *C*_4_), an expanded set of 8 channels (*F*_3_, *F*_4_, *T*_7_, *C*_3_, *C*_4_, *C_Z_*, *T*_8_, and *P_Z_*), a broader array of 18 channels (*C*_5_, *C*_3_, *C*_1_, *C*_2_, *C*_4_, *C*_6_, *CP*_5_, *CP*_3_, *CP*_1_, *CP*_2_, *CP*_4_, *CP*_6_, *P*_5_, *P*_3_, *P*_1_, *P*_2_, *P*_4_, and *P*_6_), and the full complement of 118 channels available in the dataset.

As depicted in Figure 3, a clear trend emerges: accuracy improves significantly as more channels are used for training. For instance, the minimal 3-channel configuration results in the lowest accuracies (ranging from 78.9% to 88.6%), while the comprehensive 118-channel setup achieves near-perfect performance, with accuracies consistently exceeding 99% for all subjects. Meanwhile, the 18-channel configuration also delivers strong performance, with accuracy figures ranging between 95.7% and 98.1%, trailing closely behind the 118-channel results. The improvement in accuracy when using 118 channels compared to 18 channels for training is slight, yet the training duration for the full set is six times longer than for 18 channels. Additionally, the 18-channel data are not only time-efficient but also clinically significant, as they are derived exclusively from the motor cortex region. In light of these results, the 18-channel configuration has been selected as the most efficient. This also highlights the BCINetV1 model’s resilience in handling the potential noise from the extraneous information in the 118 channels, maintaining exceptional performance.

#### 4.1.4. Statistical Analysis of Features

In this section, we conduct a statistical analysis of the extracted features for our proposed BCINetV1 model to assess the relevance of extracted hallmarks and better interpret the model. In this regard, we have considered the input EEG data from subject AA of Dataset 1 and extricated 2400 deep features with the BCINetV1 model. Figure 4 illustrates five statistical measures, namely mean, median, standard deviation, first quartile, and third quartile, respectively, for class 1 and 2 features. Such statistical metrics are significant to analyze in order to assess the central tendency measure, distribution, dispersion, and outliers in the features which are all vital for class segregation analysis. For better visualization, the statistical values of features from class 1 are arranged in ascending order, while the similar arrangement belonging to class 1 was transferred to class 2 features for fair comparison. It can be seen that the mean and median values of hallmarks from both classes are antipode to each other. This validates that the features from different classes originate from distinct distributions with well-separated central tendency measures. Similarly, the standard deviation depicts an inversely proportionate relationship between the two classes, which is yet another indicator that our model is sensitive to the variability in the data, potentially augmenting its predictive accuracy and generalization capability. Furthermore, the analysis of the first and third quartiles for the two classes reveals a consistent pattern of separation, indicating that not only the central values but also the spread and range of the data differ significantly between classes. This separation in the lower and upper quartiles underscores the model’s ability to distinguish between the classes across the entire distribution spectrum, enhancing its classification robustness and reliability.

To further evaluate the extracted features, we performed a pairwise analysis of variance (ANOVA) test for determining the statistical significance (*p*-value) of each feature pair. Figure 5a shows the correlation map of ANOVA test *p*-values, where the pairwise test is conducted for each and every combination of 2400 features. Hence the total number of tested combinations was 2,878,800. As a general practice in research, the null hypothesis cutoff threshold was set to 10% of the entire population. For better representation, those tests with *p*-values > threshold were set to 1 (non-significant white pixel), and those with *p*-values < threshold were set to 0 (significant black pixel). Consequently, we achieve a binary image with black and white pixels only. It is apparent from the Figure 5a that the figure is black dominant, clearly suggesting that a substantial proportion of the feature pairs exhibit statistically significant differences, highlighting the potential for these features to contribute meaningfully to the predictive modeling process. Next, in Figure 5b we have further quantified the occurrences of significant pairs in the form of a histogram. It is evident that the number of *p*-value instances between 0 and 0.1 is the highest (>95%), while the non-significant instances are <5% in total. The combined statistical tests incorporating mean, median, standard deviation, first and third quartile, and lastly the ANOVA test reveal that the BCINetV1 model produces statistically meaningful features which augments the class discrepancy of the MI EEG tasks and ameliorates the predictive accuracy of the model.

#### 4.1.5. Topographical Maps and Features Representation

To illustrate the significance of the features extracted by BCINetV1 for MI EEG tasks, topographical representations are provided in Figure 6. These topo-plots are generated by averaging the features extracted from each of the 18 sensorimotor cortex channels across all subjects in Dataset 1. The electrodes were methodically selected to ensure uniform coverage across both hemispheres: on the left hemisphere, the channels *C*_5_, *C*_3_, *C*_1_, *CP*_5_, *CP*_3_, *CP*_1_, *P*_1_, *P*_3_, and *P*_5_ are positioned, whereas on the right hemisphere, the channels *C*_6_, *C*_4_, *C*_2_, *CP*_6_, *CP*_4_, *CP*_2_, *P*_6_, *P*_4_, and *P*_2_ are located, demonstrating a symmetric distribution for comprehensive spatial analysis. The figure reveals a consistent pattern indicating that the features extracted by BCINetV1 effectively discriminate between hand and foot MI EEG tasks. Notably, an increase in model activations is observed in the left hemisphere of the motor cortex during hand MI tasks and conversely in the mid-right hemisphere for foot MI tasks.

This underscores BCINetV1’s robust capability to pinpoint task-specific information across the relevant channels. The topographical plot further illustrates that the intensity of activity, both positive and negative, fluctuates across different channels, suggesting that the imagined actions are detected irrespective of the spatial positioning of the channels, highlighting the model’s precision in capturing MI tasks. Furthermore, to illustrate the separability of the extracted features, we have used t-distributed stochastic neighbor embedding (t-SNE) method to plot the high-dimensional features into a two-dimensional scatter plot as shown in Figure 7. The plots are portrayed for all four datasets used in this study each having distinct number of classes. This visualization represents data from all four datasets used in our study, each with a different number of classes. Initially, the pre-trained features appear intermixed, lacking discernible patterns. However, post-training, the features segregate into distinct clusters for each class, showcasing the BCINetV1 model’s robust capability for feature separation. This evidence confirms that our model not only achieves high accuracy but also extracts clinically relevant features for the motor imagery tasks at hand.

#### 4.1.6. Ablation Study

In this section, we conduct an ablation study to assess the individual contributions of the BCINetV1 model’s components to its overall performance. Ablation studies are crucial for dissecting the model’s architecture, thereby improving its transparency, gaining deeper understanding of the learning mechanisms, and refining the parameter settings for optimal performance. Table 3 displays the classification performance and the statistical significance of features derived from various combinations of modules within the BCINetV1 framework. The accuracy figures provided are the average results for the five subjects from Dataset 1. It has been noted that excluding the ConvSAT block from the combination of T-CAB and S-CAB modules leads to diminished classification effectiveness and less significant features. This is likely because the ConvSAT block acts as a central processing element for both T-CAB and S-CAB, capturing essential local and global contexts within the EEG signals. In its absence, the T-CAB and S-CAB modules struggle to pinpoint the critical information in the signals, thereby impairing the overall classification performance.

Furthermore, it is evident that the SEB module becomes ineffective without channel reordering. This ineffectiveness is attributed to the fact that channel reordering facilitates the adjacent pairing of corresponding temporal and spectral channels, which are then efficiently processed by the group convolution layer for enhanced feature extraction. Omitting this reordering step causes a separation of relevant temporal and spectral channels, leading to the SEB block’s inability to detect shared patterns within the data. Continuing the analysis, the SEB, T-CAB, and S-CAB modules independently achieve classification accuracies of 72.45%, 75.98%, and 85.09%, respectively. This progressive improvement in accuracy reflects each module’s effectiveness in extracting MI task-relevant patterns from the EEG data. However, it is noteworthy that none of these modules reach their maximum accuracy potential when operating independently. This underlines that the true strength of the BCINetV1 model lies in the combined operation of its modules, emphasizing that an integrated approach is crucial for achieving the best results in MI task classification.

To illustrate the complete workflow of the BCINetV1 framework, Figure 8 presents a schematic diagram that traces the journey from data input to SEB output, mapping out how the T-CAB, S-CAB, and SEB modules interact within the network architecture. For better visualization, we have chosen a sample signal with apparent ERS/ERD signatures in the time-frequency domain; however, the BCINetV1 model is equally robust for a diverse ranges of MI EEG signals. It can be seen in Figure 8a that the initial 1D temporal signal does not make much sense visually, and it is impossible to make any interpretation about such a signal using clinical analysis. In Figure 8b, this signal is transformed into a 2D tempo-spectral representation, with time stamps along the x-axis and frequency on the y-axis. This transformation yields a signal that provides insightful information about spectral variations over time. However, the resulting 2D signal is marred by excessive noise and contains an overwhelming amount of extraneous information.

In Figure 8c,e, we can observe that the attention maps/masks generated by the T-CAB and S-CAB modules are localized in the time and frequency domains, respectively. The resultants in Figure 8d,f clearly showcase the precise temporal and spectral instances captured by the T-CAB and S-CAB modules by amplifying the task-related instances and attenuating the non-significant ones. Lastly, Figure 8g gives the final output as yielded by the SEB module. The SEB module successfully integrates the responses from the T-CAB and S-CAB modules, as demonstrated by the final output. This output provides a clear clinical indication of the ERS/ERD in the signal, highlighted by three red distinct spikes within the β-rhythm. This integration, combined with the spatial origin of the signal, facilitates a straightforward clinical determination of the specific MI task represented in the EEG data.

#### 4.1.7. Computational Complexity Analysis

This section delves into the computational complexity of the BCINetV1 model, focusing on how variations in input dimensions affect its processing time. As mentioned previously, the network’s input is a two-dimensional image, with time stamps along the *x*-axis and frequency on the *y*-axis. Our aim is to assess the impact of altering these dimensions on the model’s training and testing durations. Figure 9 presents this analysis, showing the changes in *x* and *y* dimensions of the input image across a range of values. We employed curve fitting techniques to accurately model and understand the computational complexity (C.C) of the BCINetV1 model under these varying conditions. Figure 9a examines how the computational time for a single-trial changes with the *x*-dimension (time stamps) of the input signal, ranging from 50 to 1000, while maintaining a constant frequency (*y*-dimension) of 100 Hz.

The findings indicate a consistent increase in both training and testing times as the number of time stamps grows. Through curve fitting analysis, a cubic polynomial, $O(n^{3})$, was determined to represent the computational complexity (C.C) in relation to the *x*-dimension increase. Conversely, Figure 9b shows a linear increase in C.C, represented by O(n), as the frequency of the input signal varies from 150 to 1000 Hz, with the time stamps held steady at 50. Consequently, the overall computational complexity of the system is established at $O(n^{3})$, which is considered reasonable for an efficient deep learning model. Additionally, the analysis revealed that the maximum time taken for processing a single trial is under 0.4 s, underscoring the model’s capability for rapid classification. In summary, the BCINetV1 model demonstrates a manageable computational complexity, scaling predictably with input size, and maintains a swift processing rate, making it a viable and efficient choice for real-time classification tasks in EEG analysis.

Figure 10 presents a multi-dimensional comparative analysis between BCINetV1 and other benchmark models illustrating the number of trainable parameters, the achieved accuracy, and the inference speed. It is important to note that the number of trainable parameters vary depending on dataset-specific characteristics like sampling rate and signal length. The values reported in Figure 10 are therefore based on the configuration for Dataset 1. However, we have confirmed that while the absolute parameter counts may shift slightly for the other datasets, the relative efficiency and the overall trend depicted here remain consistent. As illustrated in Figure 10, BCINetV1 is positioned in the top-left quadrant, signifying its ability to deliver state-of-the-art performance with a remarkably low number of trainable parameters. Specifically, BCINetV1 achieves a leading accuracy of 98.6% with only 19,648 parameters. In stark contrast, many contemporary models that achieve high accuracy, such as DeepConvNet (92.8%), EEG-Conformer (88.3%), and TCANet (93.1%), occupy the right side of the plot, requiring more parameters (164k, 158k, and 307k, respectively). This highlights a significant computational expense for a comparatively lower performance. When compared to models with a similar parameter, such as MSCNN (19,396 parameters), the BCINetV1 demonstrates a clear advantage by achieving a substantial 3% gain in accuracy. This analysis robustly demonstrates that the architectural design of BCINetV1, which integrates feature extraction and attention in a streamlined manner, successfully balances high predictive power with computational parsimony. This establishes it as a highly efficient and effective solution for MI EEG decoding, particularly for practical applications where computational resources may be limited.

#### 4.1.8. Analysis with Model Parameters

In this section, our objective is to fine-tune the primary training parameters of the BCINetV1 model to strike an optimal balance between precision and computational efficiency. The key training parameters in deep learning models include the learning rate (*lr*), the number of training epochs, and the selection of an optimal optimizer function. We aim to identify the most effective combination of these parameters for the five subjects in Dataset 1 and subsequently apply the same optimized settings to Datasets 2 through 4. Figure 11 illustrates the empirical outcomes for these parameters across various settings, providing a comprehensive view of their performance impact.

Figure 11a delves into the learning rate’s (*lr*) impact on the training accuracy for various subjects. The *x*-axis spans *lr* values from 1 $\times \;10^{-1}$ to 1 $\times \;10^{-6}$ across six logarithmic increments, illustrating how *lr* influences the speed at which the loss function minimizes and, consequently, the maximization of classification accuracy. A too high *lr* might prevent convergence, leading to poor accuracy, whereas an excessively low *lr* can decelerate the convergence process, extending the training period unnecessarily. Our analysis shows that higher initial *lr*s, specifically 1 $\times \;10^{-1}$ and 1 $\times \;10^{-2}$, do not converge, yielding low accuracy levels. A noticeable improvement in accuracy occurs at the *lr* of 1 $\times \;10^{-3}$, although it remains inconsistent across subjects. The accuracy stabilizes and maintains a consistent trend from the *lr* of 1 $\times \;10^{-4}$ to 1 $\times \;10^{-6}$, with a minor dip observed at 1 $\times \;10^{-6}$, indicating a slowdown in convergence and a need for more extended training. Consequently, based on the experimental evidence, the *lr* of 1 $\times \;10^{-4}$ was identified as optimal and was uniformly applied in subsequent experiments throughout this research.

In our second analysis, we assess how the number of training epochs affects the accuracy for each subject. Training epochs refer to the total number of complete passes the neural network makes through the entire training dataset. Insufficient epochs can lead to underfitting, while too many epochs might cause the model to overfit and unnecessarily extend the training time. The data presented in Figure 11b shows that starting from zero epochs, the model yields poor accuracies. However, there is a significant improvement after just one training pass through the dataset. The accuracy for all subjects reaches the peak after about 10 epochs, with little to no further changes. Based on these observations, we determined that 10 epochs provide an optimal balance for training the BCINetV1 model, highlighting the model’s rapid adaptability to the training data and its ability to achieve peak accuracy with minimal training time.

Finally, we explored how different optimizers affect the accuracy of our subjects. Optimizers play a crucial role in adjusting the loss function during backpropagation, and a well-chosen one can lead to quicker convergence, enhanced generalization, and improved model stability. In our study, we evaluated the performance of three widely used optimizers in EEG research: adaptive moment estimation (ADAM), root mean square propagation (RMSProp), and stochastic gradient descent with momentum (SGDM). The results, as depicted in Figure 11c, demonstrate that the ADAM optimizer markedly surpasses RMSProp and SGDM in terms of classification accuracy. ADAM’s superior performance is attributed to its effective moment estimation, bias correction, adaptive learning rates, and robustness against hyperparameter variations. Consequently, the ADAM optimizer was selected for subsequent experiments within this research, based on its standout empirical performance.

### 4.2. Analysis with Other BCI EEG Datasets

This section evaluates the performance and generalizability of BCINetV1 on three additional, distinct BCI EEG datasets. The results presented herein for each dataset represent the average performance across all subjects; however, detailed subject-wise classification outcomes are comprehensively provided in the Appendix A accompanying this manuscript. The comparative analyses in the following subsections are conducted in terms of accuracy, recall, f-score, and kappa to offer a thorough assessment of BCINetV1 against state-of-the-art methods.

As indicated by Table 4, the BCINetV1 demonstrated all-around performance on *Dataset 2*, achieving a high average accuracy of 96.6%, recall of 96.6%, an f-score of 96.5%, and a kappa score of 91.9%, all with very consistent results across the 52 subjects. When compared to hybrid signal processing methods, BCINetV1 was clearly superior; for example, its accuracy was 4.6% higher than the DR+ICA+SVM method by Tyler et al. [48] (92.0% accuracy), and it significantly surpassed the EFD-CNN method by Binwen et al. [49], being 6.7% more accurate (EFD-CNN had an accuracy and f-score of 89.9% and a kappa of 79.8%). Against the benchmark deep learning models, BCINetV1 also maintained a strong lead across all metrics. For instance, the ST-Transformer model achieved 93.7% accuracy (making BCINetV1 2.9% higher), 94.3% recall, a 93.3% f-score, and an 87.4% kappa, all of which are noticeably lower than BCINetV1’s scores. Similarly, models like TCANet (92.5% accuracy, meaning BCINetV1 was 4.1% higher; 93.7% f-score; 84.9% kappa) and EEG-Conformer (84.9% accuracy, resulting in an 11.7% lead for BCINetV1; 85.6% f-score; 69.8% kappa) lagged behind BCINetV1 in these measures. Simpler deep learning architectures such as DeepConvNet and ShallowConvNet showed even larger performance gaps across all these indicators.

For the three-class *Dataset 3* (given in Table 5), BCINetV1 demonstrated stable performance, achieving an average accuracy of 97.2%, recall of 96.9%, an f-score of 97.5%, and a kappa of 95.3%, all with a low standard deviation of 0.3%. When compared to hybrid signal processing methods, BCINetV1 was 1.1% behind CADMMI-SDI by Sadiq et al. [50] (99.3% accuracy). However, this difference is insignificant. Additionally, the CADMMI-SDI belongs to the category of hybrid signal processing methods, which has their inherent limitations mentioned in Section 1. Moving forward, the BCINetV1 shows a considerable advantage; for instance, its accuracy was 3.4% higher than the EFD-CNN method by Binwen et al. and the PCA-based RF Model by Siuly et al. [51] (83.2% accuracy, resulting in a 14% lead for BCINetV1). Against other deep learning models, BCINetV1 also performed impressively. While the EEG-Conformer matched BCINetV1’s accuracy at 97.2% (with 98.3% recall, 98.0% f-score, and 95.7% kappa), BCINetV1 achieved its results with remarkable consistency, as indicated by its extremely low standard deviation. Other strong deep learning models like TCANet (96.0% accuracy, meaning BCINetV1 was 1.2% higher; 95.5% f-score; 94.0% kappa) and ST-Transformer (94.7% accuracy, giving BCINetV1 a 2.5% advantage; 95.3% f-score; 92.1% kappa) did not reach BCINetV1’s level of combined accuracy and stability. Simpler architectures such as EEGNet (83.1% accuracy, where BCINetV1 was 14.1% higher; 83.5% f-score; 74.6% kappa) and ShallowConvNet (84.8% accuracy, showing a 12.4% lead for BCINetV1; 86.0% f-score; 77.1% kappa) showed considerably lower performance on all metrics. sensors-25-04657-t004_Table 4Table 4Comparison chart of Dataset 2 with state-of-the-art methods. To facilitate comparison, studies are ordered by increasing average classification accuracy.CategoryAuthored ByYearMethodAccuracy (%)Recall (%)F-score (%)Kappa (%)Hybird signalprocessing methodsKumar et al. [52]2019CSP+LSTM68.1 ± 9.0683.3 ± NA-65.0 ± NAKumar et al. [53]2021OPTICAL+69.5 ± NA--39.8 ± NAHossain et al. [54]2021SVM+LR+NB+KNNFFS71.1 ± 6.7770.2 ± 5.6079.0 ± 10.08-Park et al. [55]20233D-EEGNet81.3 ± 7.27---Yu et al. [25]2021EFD83.8 ± NA-83.8 ± NA-Binwen et al. [49]2022EFD-CNN89.9 ± NA-89.9 ± NA79.8 ± NAFan et al. [56]2023TFTP+3D-CNN91.9 ± NA---Tyler et al. [48]2021DR+ICA+SVM92.0 ± NA---Deep Learning methodsSimulated2025DeepConvNet67.7 ± 0.4667.7 ± 0.4667.7 ± 0.4635.3 ± 0.91Simulated2025ShallowConvNet70.7 ± 4.5470.6 ± 5.1370.0 ± 4.4041.3 ± 9.20Simulated2025FBCNet75.9 ± 2.9276.8 ± 3.0176.1 ± 2.4151.7 ± 5.86Simulated2025CCNet78.8 ± 4.8880.0 ± 5.8479.7 ± 5.6457.5 ± 9.78Simulated2025EEG-Conformer84.9 ± 2.9983.9 ± 2.4585.6 ± 3.2569.8 ± 5.98Simulated2025TSception85.3 ± 4.7684.5 ± 5.0885.0 ± 5.1669.9 ± 9.52Simulated2025MSCNN90.6 ± 2.9590.9 ± 3.5290.3 ± 2.1981.1 ± 5.92Simulated2025ATCNet91.7 ± 1.0790.9 ± 0.1191.4 ± 0.2583.3 ± 2.15Simulated2025TCANet92.5 ± 3.0092.7 ± 2.7493.7 ± 3.5984.9 ± 6.02Simulated2025ST-Transformer93.7 ± 1.7794.3 ± 1.8393.3 ± 1.5687.4 ± 3.54
**This Study**2025BCINetV196.6 ± 1.7096.6 ± 1.7396.5 ± 1.4191.9 ± 2.51

For the four-class *Dataset 4* given in Table 6, BCINetV1 again demonstrated superior performance across all metrics, achieving an average accuracy of 98.4%, recall of 98.4%, an f-score of 98.4%, and a kappa of 97.8%, with a low standard deviation of 0.6%. When compared to hybrid signal processing methods, BCINetV1’s advantage was substantial. For example, its accuracy was remarkably 15.9% higher than the ESVL method by Luo et al. [58] (82.5% accuracy; 65% kappa). Against other deep learning models, BCINetV1 also showcased a significant lead. The attention-based AMSTCNet by Zhang et al. [59] achieved an accuracy of 87.5% (88% f-score; 83% kappa), meaning BCINetV1 was 10.9% more accurate. Similarly, ATCNet by Altaheri et al. [60] (85.4% accuracy; 81% kappa) was outperformed by BCINetV1 by 13%. Even robust general benchmarks like EEG-Conformer by Song et al. [33] (78.6% accuracy; 71.55% kappa) and EEGNet (76.4% accuracy; 68.6% kappa) showed considerably lower accuracies, with BCINetV1 being 19.8% and 22% higher, respectively.

In summary, the comprehensive analysis across these diverse BCI EEG datasets underscores the stability and generalizability of BCINetV1. While many contemporary deep learning models, such as ST-Transformer which performed well on Dataset 2 (93.7% accuracy) and Dataset 3 (94.7% accuracy) but saw its accuracy drop to 57.7% on the more complex Dataset 4, or EEG-Conformer which achieved 97.2% on Dataset 3 but only 78.6% on Dataset 4, exhibit significant performance fluctuations and inconsistencies when faced with different data characteristics or task complexities. In stark contrast, BCINetV1 consistently delivered exceptionally high accuracy, recall, f-score, and kappa values across all tested datasets, regardless of the number of classes, subjects, or specific recording paradigms. This consistent, high-level performance strongly suggests that BCINetV1’s architecture effectively captures robust and discriminative features, making it a highly reliable and dataset-independent solution for MI EEG decoding.

## 5. Limitations and Future Work

The BCINetV1 model showcases promising potential for accurate and clinically significant classification of EEG signals for the realization of BCI systems; nonetheless, there is room for improvement in several areas: (1) While effectively processing the non-stationary temporal and spectral data, the model lacks a mechanism for spatial information analysis. Currently, the selection of 18 motor cortex electrodes is manual, but this process could be automated by integrating a spatial information processing block into the model. (2) The model could be further improved by adding a module for extracting common features across subjects in an end-to-end manner, making it more suitable for cross-subjects classification tasks. (3) As the name implies, the BCINetV1 is the first version of a planned series of networks. Future developments will focus on creating a more generalized framework capable of handling not only MI EEG but also other BCI paradigms like P300, ERP, and SSVEP, thereby broadening the model’s applicability and effectiveness.

## 6. Conclusions

This paper introduced BCINetV1, a new deep learning model for decoding motor imagery from EEG signals. The model uses three main components: a temporal convolution-based attention block (T-CAB), a spectral convolution-based attention block (S-CAB), and a squeeze-and-excitation block (SEB). The T-CAB and S-CAB modules, using a novel convolutional self-attention (ConvSAT) mechanism, identify important time-based and frequency-based patterns in the EEG data. The ConvSAT mechanism helps capture both local details and broader context. The SEB module then combines these identified patterns for classification. Experiments were conducted on four different MI EEG datasets, which included tasks with two or more choices. BCINetV1 consistently achieved high average classification accuracies of 98.6% (Dataset 1), 96.6% (Dataset 2), 96.9% (Dataset 3), and 98.4% (Dataset 4), along with strong results for recall, f-score, and kappa. When compared to several existing methods, BCINetV1 generally showed better accuracy and more stable performance across these diverse datasets. The study also indicated that BCINetV1 is computationally efficient, can extract clinically meaningful features, and addresses the non-stationary nature of EEG signals without needing complex pre-processing steps. In summary, BCINetV1 offers an effective and interpretable approach for MI EEG decoding, suitable for brain–computer interface applications.

## Figures and Tables

**Figure 1 sensors-25-04657-f001:**
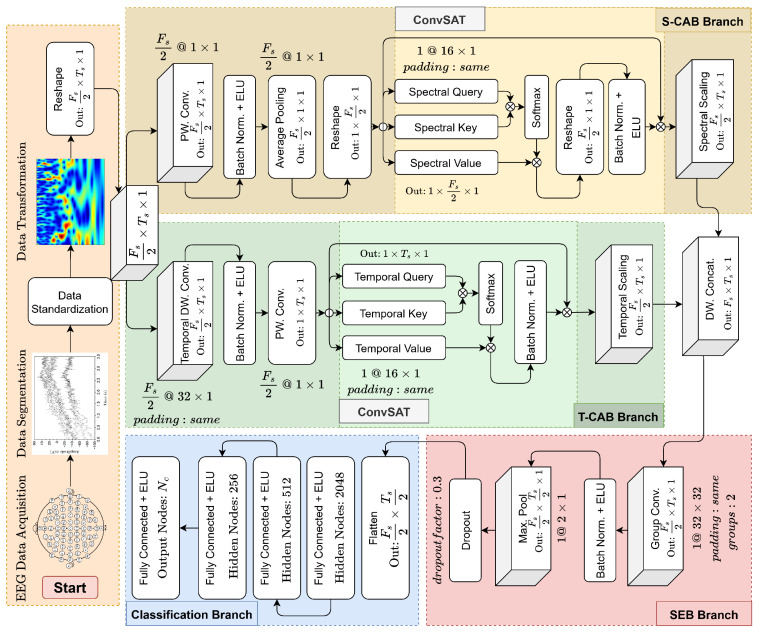
Detailed structure of BCINetV1 model with its constituent T-CAB, S-CAB, ConvSAT, and SEB blocks. Throughout the figure, the notation *k* @ m×n is used to specify a convolutional layer with *k* kernels of size m×n.

**Figure 2 sensors-25-04657-f002:**
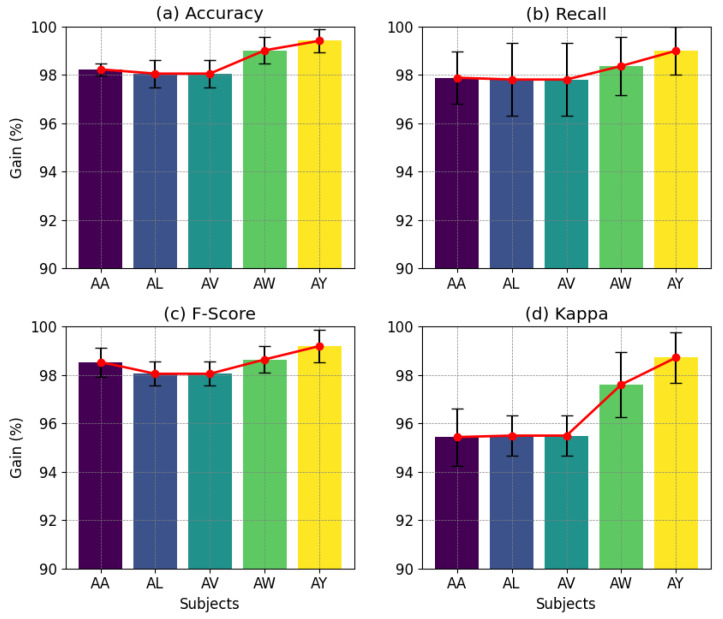
Five-fold classification performance evaluation of the BCINetV1 model, depicting (**a**) accuracy, (**b**) recall, (**c**) f-score, and (**d**) kappa across five subjects (AA, AL, AV, AW, and AY). The bar heights denote the mean outcomes while the sides whiskers represent the standard deviation.

**Figure 3 sensors-25-04657-f003:**
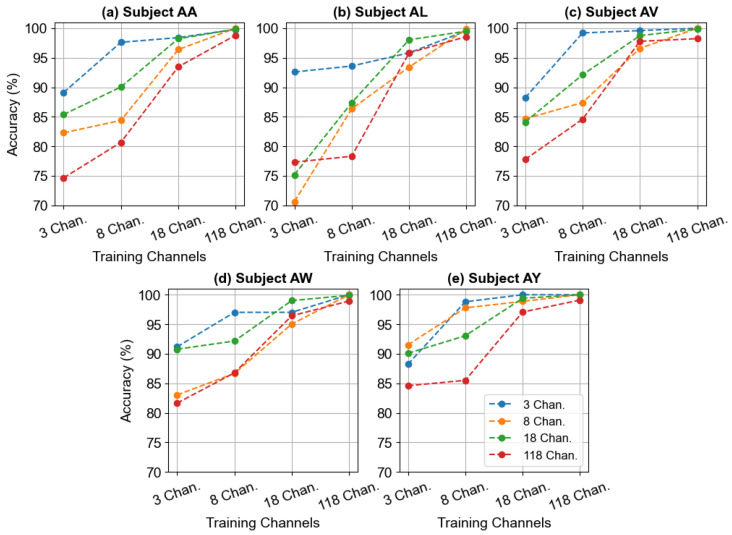
Illustration of classification accuracy for five subjects from Dataset 1 using different channel combinations. The x-axis indicates the number of electrode channels used for training (3, 8, 18, and 118 channels), while the y-axis shows the corresponding test accuracy. The channel combinations in the figure legend illustrate the channels used as the test data.

**Figure 4 sensors-25-04657-f004:**
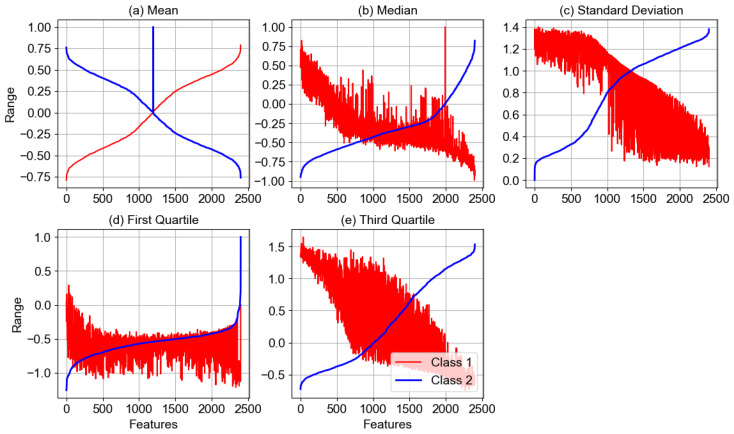
Statistical quantification of BCINetV1 for Dataset 1.

**Figure 5 sensors-25-04657-f005:**
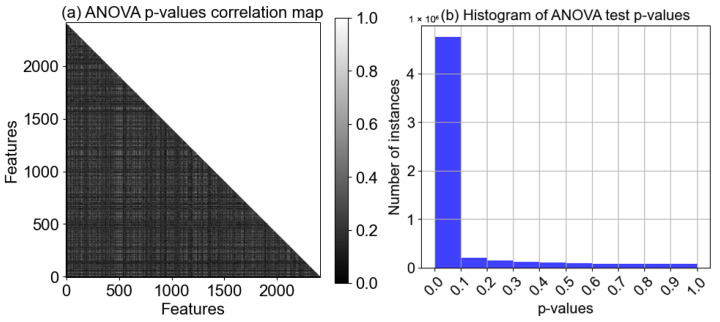
(**a**) Pairwise statistical significance (*p*-values) of features using ANOVA test. (**b**) Histogram of ANOVA *p*-values.

**Figure 6 sensors-25-04657-f006:**
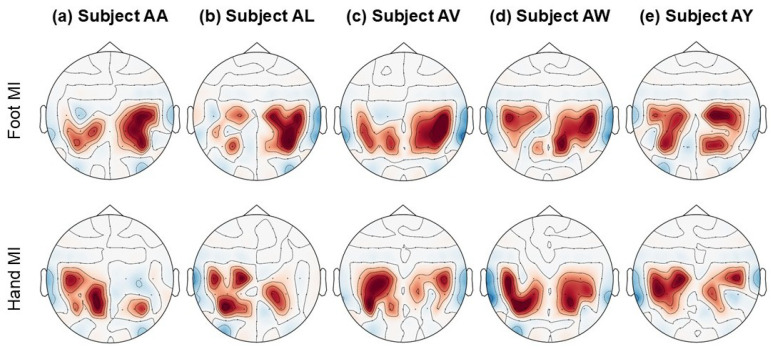
Topological maps illustrating the model activations in different regions corresponding to the imagined activity. These maps represent the average response from the SEB module, calculated over the entire 3.5 s trial duration for each task. Dark red indicates elevated activity, while the blue shades represent decreased activations.

**Figure 7 sensors-25-04657-f007:**
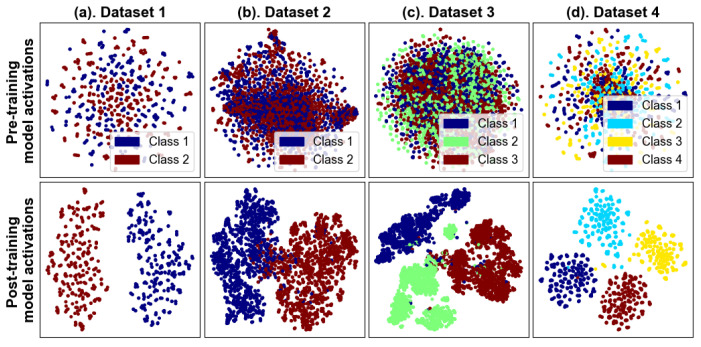
BCINetV1 2D t-SNE embeddings for feature separability in Datasets 1–4.

**Figure 8 sensors-25-04657-f008:**
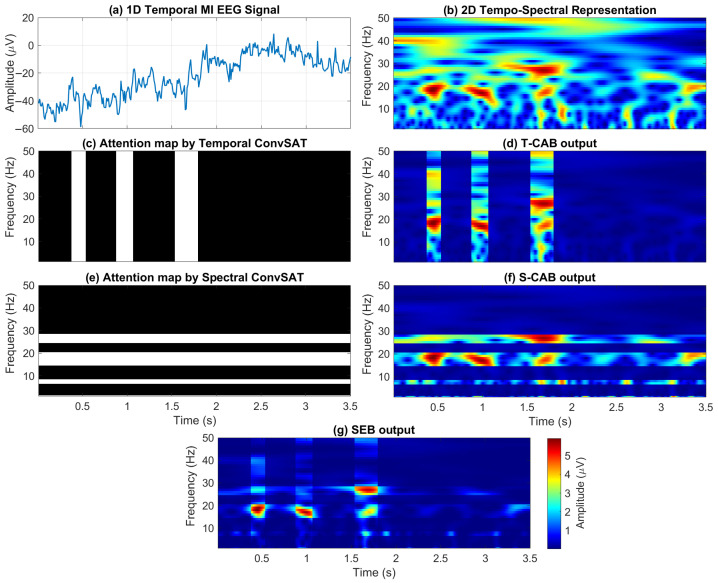
Illustration of the working mechanism of T-CAB, S-CAB, and SEB modules in BCINetV1.

**Figure 9 sensors-25-04657-f009:**
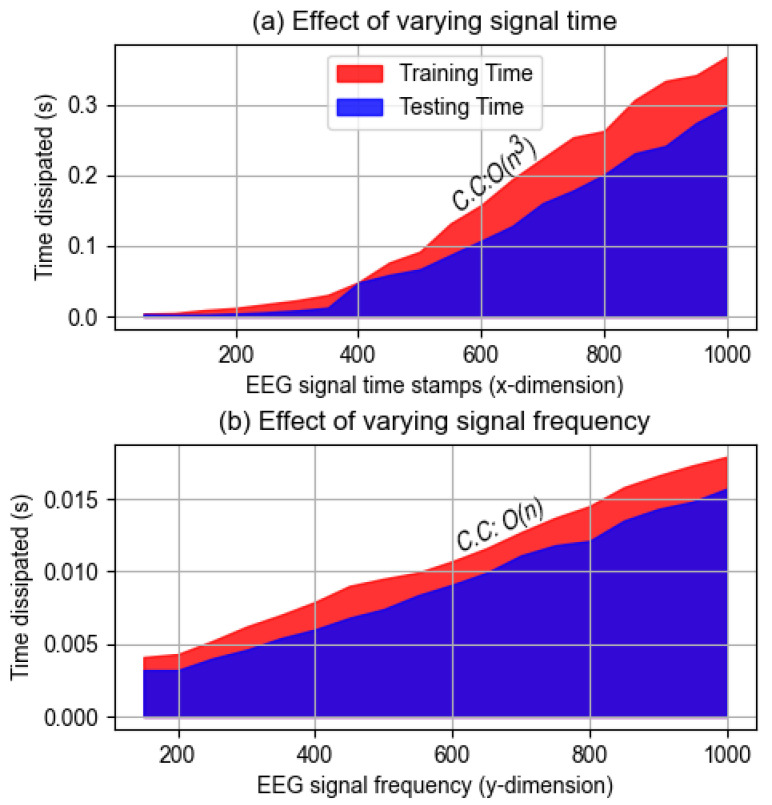
Variations in computational durations by altering the (**a**) time stamps; (**b**) frequency of the input MI EEG signals.

**Figure 10 sensors-25-04657-f010:**
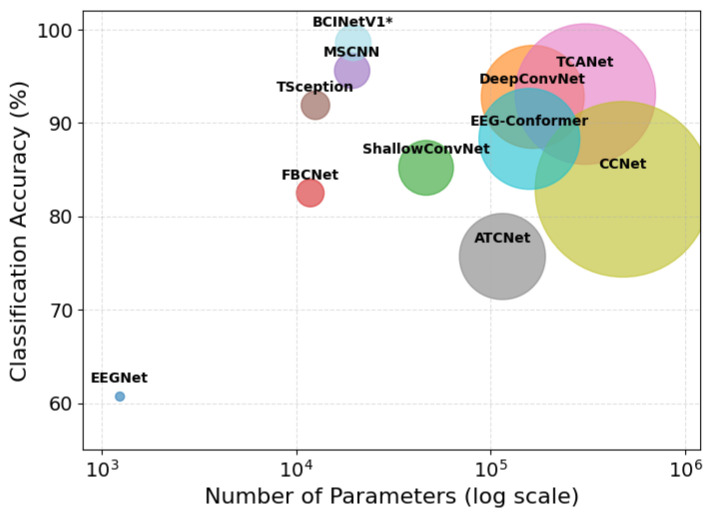
Multi-dimensional comparison of model performance on Dataset 1. Classification accuracy (*y*-axis) is plotted against the number of trainable parameters (x-axis, log scale). The size of each bubble denotes the inference time of the model. Larger bubble means higher inference time and vice versa. * denotes the highest classification accuracy model.

**Figure 11 sensors-25-04657-f011:**
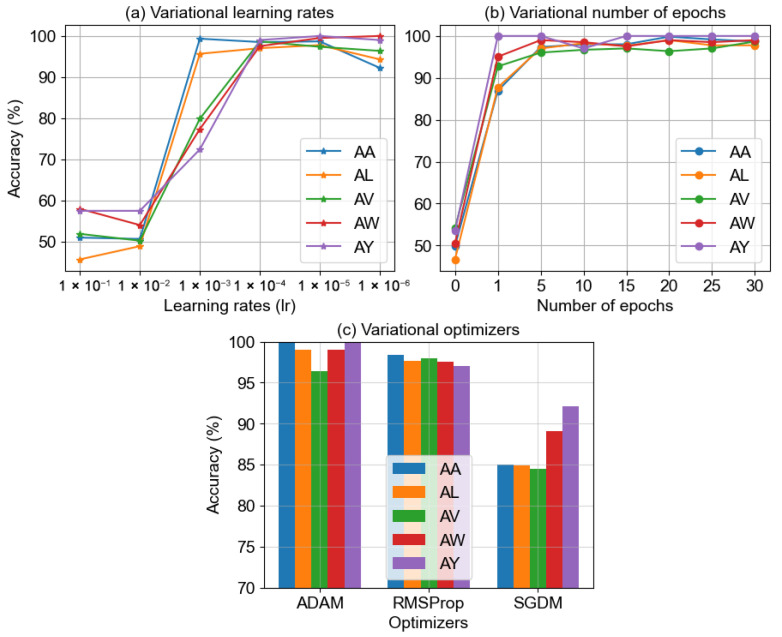
Impact of varying model hyperparameters (**a**) learning rate, (**b**) number of epochs (**c**), and optimizers over the classification outcomes.

**Table 1 sensors-25-04657-t001:** Overview of benchmark EEG decoding models for comparison.

Sr. #	Model Name	Architecture/Working Mechanism	Significance	Relevance to BCINetV1
**CNN-based Models**				
1.	EEGNet [16]	Compact; temporal convolutions for frequency filters, followed by depthwise separable convolutions for efficient spatial filtering across EEG channels. Extracts time-domain dynamic features and their spatial distributions.	Highly efficient, generalizable benchmark for various EEG paradigms, good with limited data/resources.	BCINetV1 also employs convolutional layers; EEGNet serves as a foundational compact CNN baseline.
2.	DeepConvNet [20]	Deeper architecture; standard convolutional and pooling layers for hierarchical feature learning from raw EEG. Extracts increasingly complex temporal and spatial features.	Early successful deep learning model for EEG, showed CNN capability for time-series brain data.	Represents a standard deep CNN approach; BCINetV1 builds upon and refines convolutional strategies.
3.	ShallowConvNet [20]	Simpler, shallow architecture; temporal convolution followed by a spatial convolution. Extracts fundamental temporal patterns (e.g., band-power) and spatial distributions.	Strong, simpler baseline capturing basic EEG features with good interpretability.	Offers a contrast to deeper models and highlights efficiency; BCINetV1 aims for both depth and efficiency in its convolutional components.
4.	FBCNet [27]	Integrates filter banks (frequency sub-bands) with CSP-like spatial filters learned within a CNN. Extracts frequency-specific spatial patterns.	Effectively combines traditional signal processing (filter banks, CSP) with deep learning.	BCINetV1 aims to learn spectral features directly; FBCNet provides a benchmark for hybrid spectral–spatial feature extraction.
5.	MSCNN [28]	Parallel convolutional pathways with varying kernel sizes/receptive fields for multiscale processing. Extracts features from short/long duration events and localized/broader spatial activities.	Captures richer features by considering information from different resolutions, robust to signal variations.	BCINetV1 incorporates multiscale principles in its design; MSCNN is a direct benchmark for multiscale convolutional approaches.
6.	TSception [29]	Multiscale temporal convolutions (inception-inspired) followed by spatial feature learning. Extracts diverse temporal features from various receptive fields for short/long-term dependencies.	Specialized for rich, multiscale temporal information extraction, crucial for dynamic brain states.	Aligns with BCINetV1’s focus on temporal feature extraction at multiple scales through its convolutional design.
**Attention-based Models**				
7.	TCANet [30]	TCN backbone enhanced with temporal self-attention. Extracts long-range temporal dependencies and weights important time points.	Strong capability for sequential EEG modeling, highlights critical temporal segments.	BCINetV1 uses attention (ConvSAT); TCANet benchmarks attention specifically on temporal sequences learned by TCNs.
8.	ATCNet [31]	Combines TCNs with attention across temporal, spatial, or feature dimensions. Extracts adaptively weighted temporal sequences and salient channel interactions.	Enhances TCN feature learning with dynamic, data-driven focus.	BCINetV1’s attention mechanism also aims for dynamic focus; ATCNet offers a comparison for TCNs augmented with attention.
9.	CCNet [32]	Explicitly models inter-channel correlations using specialized convolutions/graphs with attention. Extracts spatial dependencies and connectivity patterns.	Focuses on complex spatial relationships in multi-channel EEG, vital for distributed brain activity.	While BCINetV1’s primary attention is tempo-spectral, CCNet provides context for attention on spatial/channel correlations.
10.	EEG-Conformer [33]	Hybrid: CNNs for local feature extraction, then Conformer blocks (Transformer-style self-attention and convolution) for global dependencies. Extracts local, fine-grained features and global contextual relationships.	Effectively leverages CNN strengths for local patterns and Transformers for global interactions.	BCINetV1 combines convolution and attention; EEG-Conformer is a benchmark for hybrid CNN-Transformer (attention) architectures.
11.	ST-Transformer or ST-DG [34]	Transformer-based; jointly models spatial (inter-channel) and temporal dependencies using factorized/specialized attention. Extracts integrated spatio-temporal dynamics.	Explicit, unified approach to capturing complex interplay between spatial and temporal aspects.	BCINetV1’s ConvSAT addresses tempo-spectral attention; ST-Transformer benchmarks pure Transformer-based spatio-temporal attention.

**Table 2 sensors-25-04657-t002:** State-of-the-art comparison for MI EEG classification on Dataset 1, featuring BCINetV1. All performance metrics are reported in percentages (%). To facilitate comparison, studies are ordered by increasing average classification accuracy.

Category	Authored By	Year	Method	AA	AL	AV	AW	AY	Avg.	Std.
Hybird signal processing methods	Belwafi et al. [35]	2019	DSAA	69.5	96.3	60.5	70.5	78.6	81.9	14.18
	Barmpas et al. [36]	2023	Brain-wave scattering Net	78.9	92.2	60.3	86.7	85.7	80.7	11.07
	Singh et al. [37]	2019	SR-MDRM	79.4	100	73.4	89.2	88.4	86.1	10.15
	Amardeep et al. [38]	2019	R-MDRM	81.3	100	76.5	87.1	91.2	87.2	8.13
	Dai et al. [39]	2019	DTMKB	91.9	96.4	75.5	81.2	92.8	87.6	7.89
	Jaidaa et al. [40]	2019	WPD+HOS+SVM	89.6	99.3	77.9	97.5	94.3	91.7	7.66
	Amin et al. [41]	2020	DFBCSP+DSLVQ+SSVM/GRBF	93.5	98.5	81.8	93.6	96.1	92.7	5.77
	Wijaya et al. [42]	2021	LRFS+TSD	93.9	92.1	98.5	94.6	96.7	95.2	2.51
	Sadiq et al. [43]	2019	MEWT+JIA+MLP	95.0	95.0	95.0	100	100	97.0	2.70
Deep learning methods	Simulated	2025	EEGNet	61.1	70.3	55.3	57.9	58.9	60.7	5.15
	Simulated	2025	ATCNet	71.1	77.1	73.5	76.5	77.7	75.7	2.51
	Simulated	2025	CCNet	79.0	76.1	85.5	86.9	86.9	82.9	4.47
	Liu et al. [44]	2022	SACNN	91.0	92.0	77.0	77.0	79.0	83.2	6.82
	Simulated	2025	ShallowConvNet	77.5	81.8	88.3	89.8	89.9	85.2	4.80
	Simulated	2025	EEG-Conformer	89.7	86.1	86.7	87.9	90.9	88.3	1.80
	Miao et al. [45]	2020	Spatial Frequency+CNN	97.2	90.0	90.0	90.0	80.0	90.0	7.10
	Simulated	2025	TSception	93.6	89.3	93.6	89.4	93.6	91.9	2.07
	Simulated	2025	DeepConvNet	92.1	94.5	92.1	89.7	95.7	92.8	2.09
	Simulated	2025	TCANet	94.1	94.7	89.7	89.6	97.4	93.1	3.03
	Mehtiyev et al. [46]	2023	DeepEnsembleNet	96	96.6	88.7	90.6	96	93.6	3.26
	Simulated	2025	MSCNN	91.8	98.7	92.1	97.8	97.4	95.6	2.97
	Sharma et al. [47]	2023	LSTM+multi-head Attention	97.5	98.3	99.5	97.6	98.4	98.2	2.72
	**This study**	2025	BCINetV1	98.2	98	98.7	99.0	99.4	98.6	0.50

**Table 3 sensors-25-04657-t003:** Ablation study on the impact of individual and combined processing modules on BCINetV1’s classification performance. The *p*-values are derived from a one-way ANOVA test performed on the features extracted by each model configuration, assessing the statistical significance of the separability between the features of different classes. Here, (*) indicates *p* > 0.1, (**) denotes *p* < 0.1, and (***) signifies *p* < 0.05.

Experimental Blocks	Accuracy (%)	Extracted Features ANOVA Test *p*-Values
T-CAB *w/o* Temporal ConvSAT	52.06	*
SEB *w/o* channels reordering	53.09	*
S-CAB *w/o* Spectral ConvSAT	55.08	*
SEB	72.45	**
T-CAB	75.98	**
S-CAB	85.09	**
T-CAB + SEB	88.87	**
S-CAB + SEB	90.67	**
T-CAB + S-CAB + SEB	98.68	***

**Table 5 sensors-25-04657-t005:** Comparison chart of Dataset 3 with state-of-the-art methods. To facilitate comparison, studies are ordered by increasing average classification accuracy.

Category	Authored By	Year	Method	Accuracy (%)	Recall (%)	F-Score (%)	Kappa (%)
Hybird signal processing methods	Siuly et al. [51]	2017	PCA based RF Model	83.2 ± 8.33	-	-	-
	Sadiq et al. [57]	2020	20-order Matrix Determinant+FFNN	91.8 ± 2.58	-	-	-
	Binwen et al. [49]	2022	EFD-CNN	93.8 ± NA	-	93.7 ± NA	86.6 ± NA
	Sadiq et al. [50]	2020	CADMMI-SDI	99.3 ± 2.60	-	-	-
Deep Learning methods	Simulated	2025	EEGNet	83.1 ± 4.44	84.2 ± 4.83	83.5 ± 4.96	74.6 ± 6.69
	Simulated	2025	ShallowConvNet	84.8 ± 1.82	85.0 ± 0.97	86.0 ± 1.96	77.1 ± 2.74
	Simulated	2025	DeepConvNet	84.9 ± 3.73	85.8 ± 4.16	84.8 ± 3.91	77.2 ± 5.60
	Simulated	2025	CCNet	86.5 ± 6.94	86.5 ± 6.81	87.0 ± 6.61	79.8 ± 10.43
	Simulated	2025	FBCNet	87.9 ± 1.55	87.4 ± 1.63	88.7 ± 2.14	81.7 ± 2.34
	Simulated	2025	MSCNN	88.9 ± 3.12	88.7 ± 3.77	88.3 ± 3.73	83.3 ± 4.68
	Simulated	2025	TSception	91.8 ± 1.79	91.3 ± 1.13	90.8 ± 1.38	87.6 ± 2.70
	Simulated	2025	ATCNet	93.4 ± 1.80	92.5 ± 1.64	92.2 ± 2.57	90.1 ± 2.71
	Simulated	2025	ST-Transformer	94.7 ± 2.55	93.6 ± 2.12	95.3 ± 1.95	92.1 ± 3.84
	Simulated	2025	TCANet	96.0 ± 1.93	95.6 ± 1.62	95.5 ± 1.57	94.0 ± 2.91
	Simulated	2025	EEG-Comformer	97.2 ± 1.95	98.3 ± 2.63	98.0 ± 1.07	95.7 ± 2.94
	**This Study**	2025	BCINetV1	97.2 ± 0.30	96.9 ± 0.547	97.5 ± 0.52	95.3 ± 0.90

**Table 6 sensors-25-04657-t006:** Comparison chart of Dataset 4 with state-of-the-art methods. To facilitate comparison, studies are ordered by increasing average classification accuracy.

Category	Authored By	Year	Method	Accuracy (%)	Recall (%)	F-score (%)	Kappa (%)
Hybird signal processing methods	Agha et al. [61]	2016	SCSSP	63.8 ± 0.49	-	-	-
	Zhao et al. [62]	2019	WaSF+ConvNet	69 ± NA	-	-	58 ± NA
	Sakhavi et al. [63]	2018	C2CM	74.4 ± NA	-	-	65.9 ± NA
	Mahamune et al. [64]	2022	stdWC+CSP+CNN	75.0 ± 0.67	-	-	-
	Luo et al. [58]	2020	ESVL	82.5 ± NA	-	-	65 ± NA
Deep Learning methods	Liu et al. [34]	2023	ST-Transformer	57.7 ± 0.01	-	-	-
	Schirrmeister et al. [20]	2017	DeepConvNet	70.9 ± NA	-	-	-
	Schirrmeister et al. [20]	2017	ShallowConvNet	73.7 ± NA	-	-	-
	Simulated	2025	CCNet	76.7 ± NA	74.5 ± NA	75.8 ± NA	68.9 ± NA
	Simulated	2025	EEGNet	76.4 ± 14.6	-	-	68.6 ± 19.5
	Song et al. [33]	2022	EEG-Conformer	78.6 ± 0.26	-	-	71.55 ± NA
	Mane et al. [65]	2020	FBCNet	79.0 ± NA	-	-	-
	Altuwaijri et al. [66]	2022	MB-EEG-CBAM	82.8 ± 11.3	83.2 ± 11.3	83 ± 0.15	77.1 ± 11.3
	Ma et al. [67]	2022	MB-HNN	83.9 ± 9.09			78 ± 12
	Altaheri et al. [31]	2023	ATCNet	85.4 ± 9.1	-	-	81 ± 12
	Liu et al. [30]	2022	TCANet	86.8 ± 10.3	-	-	-
	Yang et al. [28]	2024	MSFCNNnet	87.1 ± 8.85	-	87 ± 9.01	82 ± 12
	Zhang et al. [59]	2023	AMSTCNet	87.5 ± 8.04	87.4±NA	88 ± NA	83 ± 0.11
	Simulated	2025	TSception	88.5 ± NA	88.5 ± NA	86.2 ± NA	84.7 ± NA
	**This Study**	2025	BCINetV1	98.4 ± 0.60	98.4 ± 0.61	98.4 ± 0.60	97.8 ± 0.81

## Data Availability

The original contributions presented in this study are included in this article. Further inquiries can be directed to the corresponding author.

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
