# Peer review of "BCINetV1: Integrating Temporal and Spectral Focus Through a Novel Convolutional Attention Architecture for MI EEG Decoding"

_sensors, 2025, doi:10.3390/s25154657_

Round 1

Reviewer 1 Report

Comments and Suggestions for Authors

This paper presents a novel deep learning architecture (BCINetV1) for Motor Imagery (MI) EEG signal decoding, achieving state-of-the-art results across four diverse datasets. The work is technically sound, well-structured, and addresses significant challenges in EEG-based BCI. There are some comments:

  1. In Tables 4, 5, 6, why do the simulation results of the same structure vary so greatly when applied to different datasets such as EEG-Conformer?
  2. As can be seen from Figure 7, there are still some overlaps among different classes. Could you analyze the reasons for this?
  3. The paper contains some minor mistakes, and the author is advised to carefully review it. (e.g. No.3 is not included in the list of the affiliations)

Reviewer 2 Report

Comments and Suggestions for Authors

The manuscript "BCINetV1: Integrating Temporal and Spectral Focus through a Novel Convolutional Attention Architecture for MI EEG Decoding" presents a new deep learning framework for motor imagery EEG classification. The authors introduce BCINetV1, which leverages a dual-branch convolutional attention mechanism—specifically, temporal and spectral convolution-based attention blocks (T-CAB and S-CAB) powered by a novel convolutional self-attention (ConvSAT) module, as well as a squeeze-and-excitation block (SEB) to integrate tempo-spectral features. The paper’s strengths lie in its clear methodological innovation, comprehensive benchmarking on four diverse datasets, and robust empirical results: BCINetV1 consistently outperforms existing state-of-the-art approaches, achieving high accuracy (up to 98.6%) and demonstrating strong generalization and stability across subjects and paradigms. The model is also computationally efficient and extracts clinically meaningful features, as evidenced by statistical and topographical analyses. The statistical analysis of the features and the ablation study are commendable.

However, some limitations remain. The current version does not address spatial information in an automated way, relying on manual selection of motor cortex electrodes (with all the electrodes achieving the best results). Overall, the work represents a significant advance in MI EEG decoding, with both practical and theoretical contributions.

Syntagm ambiguities:

    -The authors use "clinically/clinical" in several places where it may not be appropriate. The first instance is in the introduction: "clinically unexplained." The meaning of this part of the sentence is unclear to me; perhaps a better explanation would help.

Introduction:

    -Lines 37–43: The explanation of what EEG is, in a scientific paper about MI, is redundant.

    -Every mention of a dataset (as early as the introduction, and further on) should have a reference (e.g., lines 71, 78, etc.).

    -When stating the challenges, what is the manifestation of "Challenge 1" and "Challenge 2," and how exactly does your method mitigate that manifestation? For "Challenge 3," did your method really localize the event horizon? If so, it needs to be stated more clearly. There is very little discussion of ERD/S in the manuscript.

    -On line 126, can you elaborate on what is meant by "informed decision-making ability" here?

    -For the contributions of the study, I would combine points 2 and 3, and perhaps even with 1, since they are all part of the same solution. This would leave the authors with two contributions, but the first would surely be noteworthy.

Experimental Results and Discussion:

    -A significant portion of paragraphs 4.1 and 4.2, including Table 1, arguably should be presented earlier in the manuscript (methodology/second chapter, or even partially in the introduction). Here, you explain large parts of the model in the results section.

    -Table 2: Some of the results of other methods are very close to yours. Have you done any statistical analysis of these results? It would be useful to see that (same for the other datasets/other tables 4, 5, 6).

    -In line 412, you state that "motor intentions predominantly occur in the motor cortex region of the frontal lobe." Why did you subsequently use frontal electrodes for the testing of 8 channels (when reducing channel number)? Those electrodes were also not included in the further selection of 18 channels.

    -Figure 6: It would be very informative to include time points (or time span if it is an average): How long was the MI, and at which point of MI was the map created?

    -Line 566: Why such a high frequency for MI? Can you elaborate further on the rationale for this selection? Also, the size of features should be explained in detail before the results section (perhaps best when discussing data/datasets).

    -Chapter 4.4: This is a very nice and thorough comparison, but, as mentioned before, it lacks statistical analysis to further corroborate the validity of the results.

Reviewer 3 Report

Comments and Suggestions for Authors

BCINetV1: Integrating Temporal and Spectral Focus through a Novel Convolutional Attention Architecture for MI EEG Decoding

Thank you for providing this manuscript for me to review.  It is an impressive undertaking. There is clearly a tremendous amount of work that has gone into this manuscript.  The writing is excellent, although much work is needed to improve the conciseness.   That being said, I stopped in the middle of reviewing the results due to concern over bias.  I would like the authors to respond before reviewing further.

The manuscript describes the development and validation of a deep learning architecture for motor imagery EEG BCI.  The authors highlight three innovative components of BCINetV1: temporal convolution-based attention block (T-CAB), a spectral convolution-based attention block (S-CAB), both driven by a convolutional self-attention mechanism (ConvSAT).  A squeeze-and-excitation block (SEB) combines features for stable and contextually-aware classification.  Evaluated on 4 datasets with high accuracies on each.

Introduction

The statement about MI-BCI promoting neuroplasticity on page 2, Line 49 requires a citation.

Prior to the paragraph starting on Page 2, line 51, the reader does not know what constitutes “MI EEG signals”.  Are you just referring to the time series data arising from EEG recordings during a motor imagery task?  Can you clarify how EEG signals display non-linearity?  Is there a distinction between time dependency and non-stationarity?

Page 2, lines 66-68.  Is OVR-FBCSP relevant to the discussion of CNNs?  Seems an extraneous detail when just CNN-LSTM can be mentioned.

For the examples of Zhang, Dai, Deng, Ingolffson, Musallam,  and Amin, rather than the accuracies achieved, it would be helpful to know what they reported as the relative improvement (accuracy improvement) of their methods over traditional traditional machine learning algorithms.

Deng , Ingolfsson and Musallam extended upon EEGNet, which is not described.

The description of TCN does not clearly distinguish it from CNN.

Define TTG.  Related – what is the high gamma dataset?

Related to attention, do the examples of Zhang, Amin provide evidence that the addition of an attention mechanism improves the performance of an attention-less DL model?

Define FCN-TA and FCN-SA.

Challenge 3 is not clear.  Why is precise localization of ERS/ERD important?  I do not understand the phrase “…they have not successfully localized the event horizon”.  If “event horizon” is a deep learning term, please explain.

In figure 1 – it is not clear what the meaning of the @ symbol is.

The authors state that BCINetV1 avoids preprocessing, but they are introducing a number of different blocks which serve in this role:  temporal convolution attention block, spectral convolution attention block, squeeze and excitation block, convolutional self-attention block.  Is there any computational benefit?

The attention mechanism is described in the introduction, but not the concept of self-attention.  What is the distinction?

Contribution 1 encompasses contributions 2 and 3. 

BCINetV1 Overview

The first paragraph repeats what is stated from line 121 to line 149.  Consider integrating the points from section 1 into this section.

Results

The majority of sections 4.1 and 4.2 (including Table 1) should be included in the methods.  The Sensors journal requirements state “We do not have strict formatting requirements, but all manuscripts must contain the required sections: Author Information, Abstract, Keywords, Introduction, Materials & Methods, Results, Conclusions, Figures and Tables with Captions, Funding Information, Author Contributions, Conflict of Interest and other Ethics Statements. Check the Journal Instructions for Authors for more details.”

The overview of Benchmark decoding models is extremely helpful to someone who is not an expert in distinction among DL methods.  I am not the right person to judge the accuracy of these descriptions though.

Sentences that describe components of a figure (like lines 359-360) should be included in the figure caption instead.

Be concise.  Lines 364-370 are unnecessary given what is stated in lines 361-363 and figure 2.

Lines 425-430 unnecessary given that figure 3 is shown.

The legend of figure 3 should clearly explain that the colors correspond to the test channel set.

********

Because of the bold statement on line 384 “When compared to hybrid signal processing methods, BCINetV1 demonstrates a clear advantage”, I had to dig deeper into reference 33.  They report accuracies in increments of 5% and no uncertainty range, even though they reported doing 10-fold cross validation on the same 280 trials per subject.  This difference in reporting may be some of the cause of the “better consistency” reported on line 387.  Also, the authors of [33] describe using right hand vs. right foot for classification. This would be an invalid comparison against right hand vs left hand.

I looked into reference [36] as the top deep learning model competitor.  They report the results of two DL models on the Competition III Dataset IVa: LSTM and transformer.  The LSTM model did report an average accuracy of 98.2% as described, but the transformer model achieved an average performance of 99.7%.  Why is this not compared?

Given these two findings, I am left to question whether the reporting is free of significant selection bias.  Before reviewing further, the authors need to respond to the inconsistencies I’ve identified in the previous two paragraphs.  Highest classification accuracy is not a prerequisite for publication, but transparent reporting is.

Round 2

Reviewer 2 Report

Comments and Suggestions for Authors

I want to thank the authors for the thorough comments and implementation of almost all of my comments to improve their manuscript. They did great work and put a lot of effort in it, improving their manuscript significantly.

Only one thing remains unresolved from my side, and that is comment #8. Authors did the statistical analysis and had some great conclusions about it in their response to me, but, they have decided to not include these results and conclusions in the manuscript. While I understand that it might "inflate" the size of the manuscript, I would argue that it is really important to point this out, at least in a form of one paragraph (the tables about statistical significance test can be added to the supplement if authors think they are too large) which outlines the statistical significance of the results.

Reviewer 3 Report

Comments and Suggestions for Authors

Thank you for the clear and thoughtful response to my concerns.  I have reviewed the remainder of the paper and my additional comments are below.

The reasoning for not using the transformer model results is excellent, but something to this effect needs to be laid out in the paper.  The closest thing I saw that related to this was on lines 754 and 784-787.  Is it something specific about Transformer models that make them inappropriate for comparison, or just [36]’s implementation?

Lines 181-213 - There are currently two sets of bullet points 1-3 which are almost entirely overlapping in content.  I believe you can remove the second set of bullet points.

The term “Ablation” is first used in the introduction, but first described in 4.1.6.  Perhaps place this description where the term is first used.

Some additional clarification is needed for Figure 3.  How are 3 channels used in the training set, and then 118 channels used in the test set?  Also, your result does not show that there is anything special about central 18 electrodes. To show this, you would need to compare against performance with a model using 18 channels outside of the motor region.

Figure 4 – Your description of sorted class 1 values in ascending order appears correct for subplot (a), but then you start sorting in ascending order for class 2. Why?

It is not clear whether class information is used in the ANOVA analysis on page 17.  How is this an ANOVA? Is the point of this analysis to tell us that 95% of features are distinct?  From the other features in the same class or different class?  Either way, figure 5a is unhelpful.

In the analysis and presentation of figure 6, I would avoid calling the features “brain activity”.

Figure 6 should be hand vs foot, correct?  Also, you had been referring to these subjects as AA, AY…  Also, the caption should say “, with blue shades representing decreased activity”

Related to this, it is not clear that the statement of observed patterns on lines 569-570 is true.

Figure 8b – Temporospectral (or spectrotemporal).  Instead of sample number on the x-axis, use actual times.

Table 3 – p-values of what?  Against random accuracy, or some baseline model accuracy?

Figure 9 – It might be helpful to define exactly what is the size of the 2D matrix that is being input.  For increasing timestamps from 50-1000, and keeping the frequency (is this sampling frequency) at 100 Hz, you increase the window of your signal from 0.5 seconds to 10 seconds.  When you increase the sampling frequency from 150-1000Hz while maintaining a constant input window of size 50, you use 1/3 to 1/20 second worth of data.  This is why the training time is substantially shorter (b).  Or maybe I don’t understand the process.  But the main difference here is that you are operating on different sized 2D matrices, and in (b) you seem to be performing the operation on a much smaller window of data.  Is varying one dimension vs the other truly different?

Figure 10 – The size of the bubbles are directly related to the x-axis.  Showing it this way is, at best, redundant and worse, conveys a relationship in the data where there is none.

Again, this may just be my preference, but all of the text in lines 737-781 is superfluous when the table is presented.  There are very few pieces of insight (lines752-753) that add to the results in the table.
